# Zymosan-Induced Murine Peritonitis Is Associated with an Increased Sphingolipid Synthesis without Changing the Long to Very Long Chain Ceramide Ratio

**DOI:** 10.3390/ijms24032773

**Published:** 2023-02-01

**Authors:** Alix Pierron, Laurence Guzylack-Piriou, Didier Tardieu, Gilles Foucras, Philippe Guerre

**Affiliations:** Interactions Hôtes-Agents Pathogènes (IHAP), Université de Toulouse, Ecole Nationale Vétérinaire de Toulouse (ENVT), Institut National de la Recherche Agronomique et Environnement (INRAE), 31300 Toulouse, France

**Keywords:** sphingolipids, ceramides, sphingomyelins, zymosan, mouse

## Abstract

Sphingolipids are key molecules in inflammation and defense against pathogens. Their role in dectin-1/TLR2-mediated responses is, however, poorly understood. This study investigated the sphingolipidome in the peritoneal fluid, peritoneal cells, plasma, and spleens of mice after intraperitoneal injection of 0.1 mg zymosan/mouse or PBS as a control. Samples were collected at 2, 4, 8, and 16 h post-injection, using a total of 36 mice. Flow cytometry analysis of peritoneal cells and measurement of IL-6, IL-1β, and TNF-α levels in the peritoneal lavages confirmed zymosan-induced peritonitis. The concentrations of sphingoid bases, dihydroceramides, ceramides, dihydrosphingomyelins, sphingomyelins, monohexosylceramides, and lactosylceramides were increased after zymosan administration, and the effects varied with the time and the matrix measured. The greatest changes occurred in peritoneal cells, followed by peritoneal fluid, at 8 h and 4 h post-injection, respectively. Analysis of the sphingolipidome suggests that zymosan increased the de novo synthesis of sphingolipids without change in the C14–C18:C20–C26 ceramide ratio. At 16 h post-injection, glycosylceramides remained higher in treated than in control mice. A minor effect of zymosan was observed in plasma, whereas sphinganine, dihydrosphingomyelins, and monohexosylceramides were significantly increased in the spleen 16 h post-injection. The consequences of the observed changes in the sphingolipidome remain to be established.

## 1. Introduction

Sphingolipids are complex molecules involved in a wide range of biological processes including the inflammatory response and the defense against pathogens. The various pathways of sphingolipid synthesis have been elucidated (Figure 1), with all sphingolipids being metabolically interconnected and in equilibrium within cells [1]. The effects of sphingolipids vary depending on the class of sphingolipid as well as the chain length of the fatty acid incorporated in ceramides [2,3]. The sphingoid bases, notably sphingosine, are important components in the defense against pathogens, whereas sphingosine 1-phosphate is involved in lymphocyte trafficking and recruitment of immune cells [4,5]. Ceramides regulate several cellular processes including fatty acid oxidation, autophagy and apoptosis, and tissue inflammation [6]. Sphingomyelins, lactosylceramides, and complex sphingolipids are components of lipids rafts that play key roles in the binding of pathogens and signal transduction [7,8,9]. Moreover, sphingomyelins and lactosylceramides have opposite effects on arachidonic acid metabolism and inflammatory responses [10]. Furthermore, due to the absence of the C4–C5 double bond in sphinganine (Figure 1), dihydroceramides and dihydrosphingomyelins play a specific role in cell survival and lipid raft formation, different to that of the corresponding ceramides and sphingomyelins [11,12,13,14]. This complexity suggests that detailed analysis of the sphingolipidome is necessary to understand the effects of sphingolipids in inflammatory processes and immune responses [1].

Zymosan, a β-glucan polysaccharide from the cell wall of *Saccharomyces cerevisiae*, is endowed with immuno-modulating properties including immune training of the monocyte/macrophage population [16,17,18]. In mouse models, intraperitoneal injection of a low dose (0.1 to 1 mg/mice) of zymosan gives rise to a sterile peritonitis that culminates within a few hours and resolves spontaneously after 1 to 3 days [19,20]. At a high dose (10 mg/mice), zymosan induces a non-septic shock with multiple organ failure, possibly leading to death [21,22]. Zymosan is mainly recognized by the dectin-1 receptor, which is a C-type lectin receptor present on membrane lipid rafts [23,24,25,26]. Zymosan also activates TLR2 signaling, and synergistic crosstalk between dectin-1 and TLR2 has been demonstrated [27,28,29]. Binding to the receptor induces leukocyte activation and the production of inflammatory mediators, such as IL1-β, IL-6, and TNF-α, as well as activation of arachidonic acid metabolism within a few minutes/hours [25,30,31,32,33,34,35].

Administration of high doses of zymosan is lethal in mice, and sphingolipids appear to play a role in the toxicity as the administration of fumonisin B1, an inhibitor of dihydroceramides and ceramides synthase (Figure 1), protects against toxicity [21]. In another study, after administration of zymosan in the paws of mice, sphingosine 1-phosphate appeared to have a minimal role in the acute inflammatory response, and changes in ceramides concentrations in the inflamed tissue occurred after prostaglandins had increased [35]. Interestingly, the effects on ceramides appeared to vary according to the carbon chain length of the fatty acid moiety, with 18:1/18:0 and 18:1/24:1 being increased whereas 18:1/20:0 and 18:1/24:0 were less affected [35]. This observation is important because the effects of ceramides on cells vary depending on the fatty acid chain length. In some models, C16 ceramides have pro-apoptotic effects on cells, whereas C24 ceramides protect against apoptosis [36,37]. Different proportions of C16 and C24 ceramides, or C14–C18 and C20–C26 ceramides, are associated with toxicity and human diseases [38,39,40]. Moreover, in addition to the cellular increase in ceramides, the mechanism by which this increase occurs is also important in explaining ceramides-induced cell death and apoptosis [41,42]. Interestingly, studies of macrophages have found that both activation of sphingomyelinase and de novo synthesis of ceramides occurred in macrophages after TLR4 activation by LPS [43], but it has not been described whether such events occurred after zymosan administration.

Not only do ceramides need to be investigated in the course of zymosan administration; sphingomyelins, monohexosylceramides, and lactosylceramides are essential for lipid raft formation and binding to the dectin-1 receptor [7,8,9]. Consequently, disruption of sphingolipid synthesis by myriocin or fumonisin B1 blocks the phagocytosis of *Candida albicans* [44]. Moreover, the relative proportion of sphingomyelins, monohexosylceramides, lactosylceramides, and ceramides in membrane rafts is important for the binding of pathogens and for signal transduction [9,45]. Additionally, lactosylceramides, ceramides, and sphingomyelins interact with cytosolic phospholipase A2 (cPLA2), and the consequences of these interactions vary depending on the class of sphingolipid; lactosylceramides and ceramides are activators of cPLA2, whereas sphingomyelins inhibit its activity [10,46]. These findings suggest that the effect of zymosan on the sphingolipidome could play a key role in the pro-inflammatory and pro-resolution phases of the inflammatory response.

The objective of this study was to measure the changes in the sphingolipidome during the first 16 h after intraperitoneal administration of zymosan in mice. The dose of zymosan used (0.1 mg/mouse) was selected to permit spontaneous recovery of peritonitis so as to avoid interference with sphingolipid production due to the cellular damage that occurs at high doses. Sphingolipid levels were determined using targeted UHPLC-MS/MS analysis [47], which allows the precise quantitation of analytes necessary to investigate the de novo and salvage pathways (Figure 1). Sphingolipid levels were measured in the peritoneal liquid obtained by lavage of the peritoneal cavity, the peritoneal cells, the plasma, and the spleenocytes in order to distinguish local from systemic effects. Cell populations and pro-inflammatory cytokines were measured in the peritoneal lavage to confirm peritonitis.

## 2. Results

### 2.1. Peritoneal Cells, Splenocytes, and Cytokines

No mortality and no macroscopic alteration at necropsy were observed in this study. Measurements of cells and cytokines were performed to confirm that the dose of zymosan administered was responsible for a low grade non-infectious inflammatory peritonitis. Cell numbers in the spleen was similar between zymosan-injected and control (PBS) mice. By contrast, a significant 4-fold increase in the total number of peritoneal cells was observed 8 h post-zymosan injection; the values then decreased, but differences among groups were not significant (Table 1). This increase was accompanied by a significant increase in CD45^+^ peritoneal cells observed at 16 h post-zymosan injection. Characterisation of the cell populations revealed that neutrophils (Ly6G^+^), and resident macrophages (F4/80^+ high^) subsets were the most abundant, with CD11c^+^ dendritic cells representing less than 1% of the leukocytes CD45^+^ cells. Changes in Ly6G^+^ paralleled those in CD45^+^, while F4/80^+ high^ decreased significantly at 2, 4, and 8 h after zymosan administration and then tended to increase again at 16 h.

The concentrations of IL-6, IL1-β, and TNF-α in the peritoneal fluid and IL-6 in the plasma are reported in Figure 2. The greatest increase was observed for IL6, and it occurred 2 h post-injection. The increase was 32-fold in the peritoneal fluid and 8-fold in the plasma. The highest concentrations of IL-1β and TNF-α were observed at 2 and 4 h post-injection, respectively. The cytokine concentrations in the peritoneal fluid and the plasma did not differ from those of the controls at 16 h post-zymosan injection.

### 2.2. Sphingolipids by Class

As the effects on sphingolipid metabolism are complex, a first analysis was carried out by classes. Sphinganine and sphingosine, and dihydroceramides, ceramides, dihydrosphingomyelins, sphingomyelins, monohexosylceramides, and lactosylceramides concentrations expressed as the sum of the sphingolipids measured per class are reported in Table 2 for the peritoneal fluid, the peritoneal cells, the plasma, and the spleen. The concentrations of the different analytes measured in these matrices are reported in Appendix A.

Intraperitoneal injection of zymosan led to a pronounced and significant change in most of the sphingolipids measured in the various matrices, and the effects varied with time. Due to the complexity of the results, the changes in sphingolipids observed at the different times post-zymosan injection are also reported in Figure 3 as percentages of the values measured in controls. Sphinganine, dihydrosphingomyelin, sphingomyelin, monohexosylceramides, and lactosylceramides were greatly increased in the peritoneal fluid at 4 h post-injection, and most of these increases were significant (Figure 3, Table 2). As shown in Figure 3, pronounced increases in all sphingolipids were also observed in peritoneal cells at 8 h post-injection, and all of these changes were significant. At this time point, a three-fold increase in total ceramides was observed, whereas there was a 13-fold increase in dihydrosphingomyelins (Table 2). The concentrations of monohexosylceramides and lactosylceramides were still above the values measured in the controls at 16 h post-injection. The effects of zymosan on sphingolipids in plasma were less pronounced and were dominated by an increase in dihydroceramides and ceramides 2h post-injection, a decrease in sphingoid bases at 8 h post-injection, and an increase in monohexosylceramides at 16 h post-injection (Figure 3, Table 2). Concerning the spleen, most of the differences measured between control mice that received PBS and mice that received zymosan were found at 16 h post-injection. These effects corresponded to an increase in sphinganine, dihydrosphingomyelins, and monohexosylceramides (Figure 3, Table 2).

Thus, the analysis of the sphingolipid classes reveals that the local effects of zymosan are much stronger than its systemic effects. As all classes of sphingolipids were significantly altered after zymosan administration (Table 2), a further presentation of the results was carried out to compare the effects on sphinganine-based sphingolipids and on sphingosine-based sphingolipids. The sphinganine to sphingosine (Sa:So), dihydroceramides to ceramides (DHCer:Cer), and dihydrosphingomyelins to sphingomyelins (DHSM:SM) ratios were calculated to reveal whether the change in sphingolipids in peritoneal fluid and in peritoneal cells after zymosan administration could be related to activation of the salvage or the de novo pathways.

As shown in Figure 4, a significant increase in Sa:So was observed at 2 and 4 h post-injection in peritoneal fluid in comparison to control, after which this ratio tended to decrease at 8 and 16h post-injection. The DHCer:Cer ratio was difficult to interpret because of a pronounced decrease that occurred at 4 h post-injection. The DHSM:SM ratio was increased in peritoneal fluid at 2 and 8 h post-injection. Concerning the peritoneal cells, the DHCer:Cer ratio was increased at 2 and 8 h post-injection, and the DHSM:SM ratio was increased at all of the post-injection time points (Figure 4). A significant increase in the DHCer:Cer ratio was also observed in plasma 8 h post-injection and in splenocytes at 2 and 4 h post-injection. Significant differences between groups were observed in the Sa:So ratio in splenocytes, with no difference from controls (Figure 4). The increase in the Sa:So, DHCer:Cer, and DHSM:SM ratios therefore suggests that the effects of zymosan are more pronounced on dihydrosphingolipids than on sphingolipids.

Because ceramides are used to form sphingomyelins, monohexosylceramides, and lactosylceramides (Figure 1), sphingomyelins to ceramides (SM:Cer), monohexosylceramides to ceramides (HexCer:Cer), and lactosylceramides to ceramides (LacCer:Cer) ratios were also measured in the peritoneal cells (Appendix A). The SM:Cer ratio remained constant with time in the four matrices analysed apart from a small increase in the spleen at 4h. By contrast, both the HexCer:Cer and the LacCer:Cer ratio were significantly increased at 16 h post-injection in the peritoneal cells and the plasma. These ratios did not vary significantly in the peritoneal fluid and were fluctuating in the spleen.

Finally, intraperitoneal administration of zymosan resulted in an increase in sphingolipids in the peritoneal fluid, reaching a maximum at 4 h post-injection, and it then tended to return to values similar to the control values. The effects of zymosan on sphingolipids in the peritoneal cells were most pronounced, and mainly observed at 8 h post-injection. Minor changes in sphingolipids occurred in the spleen, and these were observed at 16 h post-injection. Comparison of the Sa:So, DHCer:Cer, and DHSM:SM ratios shows that the effects of zymosan are more pronounced on sphinganine-containing sphingolipids than on sphingosine-containing ones.

### 2.3. Sphingolipids According to the Chain Length of the Fatty Acid

The comparison of the effects of zymosan on different classes of sphingolipids does not allow us to determine whether different sphingolipids within the same class are affected in the same way. This information is nevertheless essential given the role of fatty acid chain length on sphingolipid. Given the number of sphingolipids measured, it is difficult to examine the effect of zymosan on an analyte-by-analyte basis. Ceramide ratios and sphingolipid correlations were therefore performed to reveal whether zymosan varied the respective proportions of the compounds measured.

The effect of zymosan according to the fatty acid carbon chain length was first investigated by measuring different ceramides ratios in the peritoneal fluid, peritoneal cells, plasma, and the spleen (Table 3). The C16 to C24 ceramide ratio (C16:C24) corresponded to the ratio among 18:1/16:0 and the sum of 18:1/24:0, 18:1/24:1, and 18:1/24:2. The C16:C24 ratio varied slightly with the time after zymosan injection, but this difference was not significant. The same was observed for the C14–C18 to C20–C26 ceramide ratio (C14–C18:C20–C26), which corresponded to the ratio between the sum of ceramides for which the fatty acid was 14 to 18 carbons divided by the sum of ceramides for which the fatty acid was 20 to 26 carbons.

The correlations between the different sphingolipids were also explored to reveal whether the administration of zymosan changed the respective ratios between compounds. Correlations among sphingoid bases and sphingolipids with fatty acids from 14 to 24 carbons were noted in peritoneal cells in the controls (Figure 5) and in mice that received zymosan, combining all the time of exposure (Figure 6). The numerical values of the Pearson coefficients of correlation observed are reported in Appendix A.

Sphingolipids of the same class with similar fatty acid chain lengths were generally positively correlated, suggesting a common regulation of these compounds. In controls not exposed to zymosan, the correlation between dihydroceramides and other sphingolipids assayed was weak or negative (Figure 5). Similarly, some ceramides, particularly those with polyunsaturated fatty acids (18:1/22:2 and 18:1/24:2), were only weakly correlated with other sphingolipids. Administration of zymosan generally increased the power of correlations between sphingolipids with the exception of 18:1/2:0, which was negatively correlated with other sphingolipids, and SM18:1/14:0, for which no significant correlation was observed (Figure 6). Notably, dihydroceramides are positively correlated with all other sphingolipids measured after zymosan injection, which was not the case in controls.

Therefore, the effects of zymosan on sphingolipids are similar regardless of the length of the fatty acid incorporated in their synthesis. Analysis of the correlations observed between the different analytes measured also revealed that zymosan administration induced a strong positive correlation between dihydroceramides and other sphingolipids, which was not observed in control mice.

## 3. Discussion

The low dose of 0.1 mg zymosan/mouse used in this study was selected to induce a low inflammatory response to allow detecting changes in sphingolipids in the absence of systemic organ failure [19,20,21,22]. The effects of zymosan on the peritoneal cells varied with the subpopulation studied. F4/80^+ high^, which corresponded to resident macrophages, decreased in the peritoneal cavity then increased in accordance with the data in the literature [48,49,50,51,52,53]. Ly6G^+^ cells, corresponding to neutrophils, increased with time post-injection, which is in concordance with previous results [49,50,53].

The pro-inflammatory cytokines IL-6, IL-1β, and TNFα increased rapidly after the injection of zymosan. The most rapid and pronounced effect was observed for IL-6, followed by IL-1β, whereas the effects on TNFα were less pronounced. The effects of zymosan on IL-6 concentrations were more marked in the peritoneal fluid than the plasma. All of these results are in keeping with the data in the literature [49,51,52]. Consequently, the effects of zymosan on peritoneal cells and on cytokines in this study are consistent with the occurrence of a mild and transient sterile peritonitis [19,49,50,52]. No effect of zymosan on the number of cells in the spleen was observed, which is consistent with the low dose that was used [21,22].

The sphingoid bases, dihydroceramides, ceramides, dihydrosphingomyelins, sphingomyelins, monohexosylceramides, and lactosylceramides varied after zymosan injection, and the effects on sphingolipids differed with the matrices measured and the analysis times. The increase in ceramides observed in peritoneal fluid and peritoneal cells in this study at 4 and 8 h post-injection of zymosan is consistent with previous data obtained by administration of zymosan in the paws of mice [35]. De novo synthesis and salvage pathway could explain increases in ceramides and sphingosine in cells (Figure 1). Sphingomyelinases are enzymes found in pathogens and present in the hosts that hydrolyse sphingomyelins, leading to an increase in ceramides [54]. Bacterial sphingomyelinases are virulence factors that are not present in zymosan, which is a polysaccharide purified from the *Saccharomyces cerevisiae* cell wall [55,56,57]. Increased concentrations of ceramides in cells secondary to activation of host sphingomyelinases have been implicated in the fusion of phagosomes and lysosomes as well as TNFα activation [54,58]. However, because an increase, rather than a decrease, in sphingomyelins was observed in this study, involvement of sphingomyelinase appears unlikely. Furthermore, in addition to ceramides, sphinganine, dihydroceramides, and dihydrosphingomyelins were also increased. Although ceramides and sphingosine can be formed by the salvage or the de novo pathways, sphinganine, dihydroceramides, and dihydrosphingomyelins can only be obtained by de novo synthesis of sphingolipids (Figure 1) [1]. Moreover, although glucosylceramidases can form ceramides in cells [59], the abundance of monohexosylceramides is low compared to sphingomyelins, and an increase, rather than a decrease, in monohexosylceramides was observed here. Dihydrosphingomyelins were the sphingolipids that were increased the most in this study. This result can be explained by the fact that dihydrosphingomyelins are end products in sphingolipid metabolism whereas sphinganine is consumed in cells to form dihydroceramides, and dihydroceramides are used to form both dihydrosphingomyelins and ceramides (Figure 1). For a long time, dihydroceramides were considered to have weak biological properties, but they are now recognized to play a role in autophagy and cytokine production [11]. Inhibition of dihydroceramide desaturase 1 exacerbated the reduced mitochondrial respiration due to LPS in cultured myotubes [60]. Although it is still difficult to conclude whether increased concentrations of DHCer are deleterious or not [61], the high concentration of dihydrosphingomyelins observed in the peritoneal cells in this study can be considered to be an adaptive response of the cells to reduce the content of dihydroceramides. Very little is known about dihydrosphingomyelins compared to sphingomyelins, but it has been suggested that dihydrosphingomyelins form ordered domains more effectively than sphingomyelins due to the double bond at C4–C5 in sphingosine [62], which can modify the repartition of some membrane proteins preferentially localized to higher-ordered dihydrosphingomyelins lipid rafts [13].

Because the effects of zymosan on ceramides, sphingomyelins, monohexosylceramides, and lactosylceramides appeared to vary, the relative abundances of these analytes in the peritoneal cells were compared. The HexCer:Cer and LacCer:Cer ratios were significantly increased at 16 h post-injection, whereas the SM:Cer ratio was unaffected. By contributing to the formation of lipid rafts, monohexosylceramides, lactosylceramides, sphingomyelins, and ceramides are essential for the binding of pathogens, polysaccharides, and other entities [7,24]. In keeping with this, myriocin and fumonisin B1, which are sphingolipid disruptors, block phagocytosis and increase the sensitivity of mice to *Candida albicans* infection [44]. Similar results have been reported for phagocytosis of *Mycobacterium tuberculosis*, which is another dectin-1-binding microorganism [63]. Changes in sphingolipid compositions in lipid rafts are known to modify the binding of pathogens to their receptor and downstream signal transduction [9,45]. Furthermore, zymosan has been found to directly bind to lactosylceramides [64]. Lactosylceramides and sphingomyelins are also known to interact with cytosolic PLA2, and the consequences of these interactions differed. Lactosylceramides are activators of cytosolic PLA2, leading to the release of arachidonic acid and the production of prostaglandins, whereas sphingomyelins inhibit the activity of cytosolic PLA2 and the release of arachidonic acid [10,46]. Consequently, the late changes in sphingolipids observed at 16 h post-zymosan injection in this study could have delayed effects on the pro-inflammatory and pro-resolution phases of the inflammatory response, which warrants further investigation.

All the ceramides measured in this study correlated together, irrespective of their fatty acid chain length. Furthermore, both the C16:C24 and the C14–C18:C20–C26 ceramide ratios remained close in controls and in treated mice. This observation is important because C16 and C24 ceramides have different apoptotic effects on cells [36,37], and changes in the C14–C18:C20–C26 ceramide ratio are associated with different diseases in humans, notably cardiovascular diseases [38,39]. Moreover, increased C18 ceramide concentrations have been found in macrophages of obese mice, suggesting that a change in ceramides ratios could have consequences for developing asthma [65]. A previous study involving zymosan administration in the paws of mice suggested that 18:1/18:0 and 18:1/24:1 accumulated more than 18:1/20:0 and 18:1/24:0 [35]. By contrast, 18:0/16:0 and 18:1/16:0 were increased in macrophages after LPS stimulation, whereas 18:1/14:0, 18:1/18:0, 18:1/20:0, 18:1/22:0, and 18:1/24:0 were not [66]. Variation in the relative abundances of ceramides according to their carbon chain length would involve a change in the expression or activity of ceramide synthases (CerS), which vary in their chain length specificities (Figure 1) [67]. For CerS, CerS2, which catalyses the formation of C20–C26 ceramides, plays a key role in liver homeostasis, and its inhibition by fumonisin B1 could explain differences in sensitivity to the mycotoxin [15,68]. Studies involving macrophages exposed to fumonisin B1 have also revealed a key role of CerS2 in the accumulation of ceramides and phagocytosis [59,69]. Furthermore, CerS2-null mice are more sensitive than wild-type mice to LPS-mediated shock, and this difference appears to be related to increased secretion of TNFα [70]. However, most of the studies conducted to date that reported an increase in de novo synthesis of ceramides in macrophages did not reveal chain length-specific effects [71,72,73], and the results obtained in this study are in concordance with these observations.

Changes in sphingolipids after injection of zymosan were observed in nearly all the matrices measured in this study, but the amplitude and the time at which these effects occurred varied. An increase in the Sa:So ratio was observed in the peritoneal fluid as early as 2 h post-injection. In peritoneal cells, the maximum effect of zymosan on dihydroceramides, including sphinganine, occurred at 8 h post-injection. These observations are consistent with the activation of de novo synthesis of sphingolipids observed after exposure of macrophages in culture to Kdo2-lipid A and lipopolysaccharide used as TLR4 ligands [71,73]. Additionally, the maximum expression of serine palmitoyltransferase and dihydroceramide synthase, which are key enzymes of de novo synthesis of sphingolipids (Figure 1), was observed at 2 and 4 h post-injection in a lipopolysaccharide murine model of lung injury [74]. Kdo2-lipid A and lipopolysaccharide are selective TLR4 agonists, whereas zymosan binds to dectin-1 receptor and activates TLR2 on membrane lipid rafts [23,24,25,26,27,28]. Binding of zymosan to dectin-1 enhances the TLR2-mediated pro-inflammatory response in macrophages, nuclear factor-kappa B activation, and cytokine production [27,28]. Although there are no data in the literature on enhancement of the de novo synthesis of sphingolipids following TLR2/dectin-1 activation, this mechanism has been observed after TLR4 activation [71,73], and a unifying model of how metabolism regulates macrophage inflammatory responses has been proposed [43,75]. Interestingly, a recent study comparing the consequences of local paw inflammation in mice due to zymosan or lipopolysaccharide revealed increased lipid concentrations at the site of inflammation in the two models, with monohexosylceramide being important for the changes observed [76]. Thus, the observed effects of zymosan on glycosylceramides in this study are consistent with the hypothesis of a unifying model of TLR2/TLR4 regulation of sphingolipid metabolism. However, it is important to note that increased de novo synthesis of sphingolipids is not the only mechanism involved in sphingolipid changes during TLR4 activation, with activation of sphingomyelinase also being reported [66,74,77,78]. Furthermore, the binding of TNFα to its receptors activates sphingomyelinase in cells [54,58], but as discussed above, this mechanism appears unlikely to have occurred in this study.

In conclusion, this study demonstrated for the first time that intraperitoneal administration of zymosan in mice leads to pronounced increases in sphingolipids. Targeted analysis suggests that the changes observed corresponded to an increase in de novo synthesis, which began 2 h after injection to reach a maximum at 8 h post-injection in peritoneal cells. No changes in the C16:C24 and C14–C18:C20–C26 ratios were observed. Monohexosylceramides and lactosylceramides were greatly increased in peritoneal cells and remained higher than in controls at 16 h post-injection. Further studies are necessary at the protein level to clarify the mechanisms involved in the changes observed and investigate their systemic consequences on the pro-inflammatory and pro-resolution phases of the inflammatory response.

## 4. Materials and Methods

### 4.1. Analytes and Reagents

FcR Blocking Reagent, Viobility™ 488/520 Fixable Dye, Ly-6G VioBlue^®^ (REAfinity™, clone REA526), CD45 VioGreen^®^ (REAfinity™, clone REA737), CD11c PE-Vio^®^ 770 (clone REA754), and F4/80 APC (clone REA126) were purchased from Miltenyi Biotec (Bergisch Gladbach, Germany). Zymosan was obtained from InvivoGen (Toulouse, France). All other analytes and reagents were purchased from Sharlab (Sharlab S.L., Sentmenat, Spain), or Sigma-Aldrich (Sigma-Aldrich Chemical Co., Saint Quentin Fallavier, France). All the reagents were HPLC analytical grade, except those used for UHPLC-MSMS, which were of LC-MS grade. The 10 sphingolipids used as internal standards obtained from Sigma-Aldrich were manufactured by Avanti Polar Lipids under the trade name “Ceramide/Sphingoid Internal Standard Mixture I” and corresponded to C17 sphingosine (d17:1), C17 sphinganine (d17:0), C17 sphingosine-1-P (d17:1P), C17 sphinganine-1-P (d17:0P), lactosyl (ß) C12 ceramide (Lac18:1/12:0), C12 sphingomyelin (SM18:1/12:0), glucosyl (ß) C12 ceramide (Glu18:1/12:0), 12:0 ceramide (18:1/12:0), 12:0 ceramide-1-P (18:1/12:0P), and 25:0 ceramide (18:1/25:0) in ethanol solution. The 33 sphingolipids used as standards obtained from Sigma-Aldrich were solubilized in ethanol at a concentration of 25 µM and corresponded to deoxysphingosine (dSo = m18:1); deoxysphinganine (dSa = m18:0); sphingosine (So = d18:1); sphinganine (Sa = d18:0); sphingosine-1-P (d18:1P); sphinganine-1-P (d18:0P); glucosylsphingosine (GluSo); lysosphingomyelin (LysoSM); lactosylsphingosine (LacSo); N-acetylsphingosine (18:1/2:0); N-acetylsphinganine (18:0/2:0); ceramides: 18:1/14:0, 18:1/16:0, 18:1/18:0, 18:1/20:0, 18:1/22:0, 18:1/24:1, and 18:1/24:0; ceramide-1P: 18:1/16:0P; dihydroceramides: 18:0/16:0, and 18:0/24:0; glucosylceramides: Glu18:1/16:0 and Glu18:1/24:1; lactosylceramides: Lac18:1/16:0 and Lac18:1/24:1; sphingomyelins: SM18:1/14:0, SM18:1/16:0, SM18:1/18:0, SM18:1/18:1, SM18:1/20:0, SM18:1/22:0, SM18:1/24:1, and SM18:1/24:0.

### 4.2. Animal and Experimental Design

Male C57BL/6 mice aged 2–3 months were bred and housed in a specific pathogen-free facility (INSERM US 006-CEFRE). The experiments were performed in an accredited research animal facility of the UMR IHAP, ENVT, Toulouse, France. Mice were handled and cared for according to the ethical guidelines of our institution and following the Guide for the Care and Use of Laboratory Animals (National Research Council, 1996) and the European Directive EEC/86/609, under the supervision of authorized investigators. Mice were euthanized by cervical dislocation, and all efforts were made to minimize animal pain and distress. Four experiments were carried out, involving a total of 36 male mice of 2–3 months of age at the start of the experiments. The animals were weighed, and peritonitis was induced by an intraperitoneal injection of 0.1 mg zymosan per mouse in a volume of 100 µL/mouse. Control animals only received sterile PBS. The animals were euthanized by cervical dislocation at 2, 4, 8, and 16 h post-injection prior to sample collection.

### 4.3. Sample Collection and Cell Isolation

Blood was collected at the cardiac level in tubes precoated with 0.5 M EDTA and then centrifuged for 10 min at 1500× *g* to harvest plasma. The plasma was stored at −80 °C until the sphingolipid analysis. Cells from the peritoneal cavity of mice were harvested by lavage with 4 mL of cold PBS. The intraperitoneal fluid was centrifuged for 7 min at 500× *g*, and the supernatant was collected and stored at −80 °C until analysis. The cell numbers were determined by flow cytometry using a MACSQuant^®^ Analyzer from Miltenyi Biotec. The cells were then suspended at 10 × 10^6^/mL in sterile PBS, and an aliquot was used for cell phenotyping by flow cytometry while the rest was centrifuged for 7 min at 500× *g*. The pellet obtained by centrifugation was recovered and a 70 µL sample was stored at −80 °C for sphingolipid analysis. Spleens were removed, and cells were isolated through a 40 μm cell strainer followed by aliquoting for counting and analysis of sphingolipids as described for the peritoneal cells.

### 4.4. Flow Cytometry Staining for Cell Counting and Phenotyping and Cytokines Measurements by ELISA

Cell numbers from the peritoneal cavity and spleen were determined by the flow cytometry absolute counting system with a MACSQuant^®^ Analyzer. Cells (0.5 × 10^6^) were incubated in HBSS, 0.5% BSA, 10 mM HEPES containing mouse FcR Blocking Reagent, following the manufacturer’s instructions. Cell viability was assessed using Viobility™ 488/520 Fixable Dye. Incubation with antibodies was performed at 4 °C for 30 min in the dark. The antibodies used were Ly-6G VioBlue^®^, CD45 VioGreen^®^, CD11c PE-Vio^®^ 770, and F4/80 APC. Acquisition was performed using a MACSQuant^®^ Analyzer flow cytometer using MACS Quantify software (Miltenyi Biotec). The flow cytometry data were analyzed using FlowJo™ Tree Star software (Ashland, OR, USA). The concentration of the different cytokines in the intraperitoneal fluid was determined using an ELISA kit from Bio-Techne (Minneapolis, MN, USA) for IL6, IL1β, and TNFα, following the manufacturer’s instructions.

### 4.5. Chromatographic System and Analysis of Standards

The sphingolipids were measured using an UPLC MS/MS system composed of a 1260 binary pump, an autosampler, and an Agilent 6410 Triple Quadrupole Spectrometer (Agilent, Santa Clara, CA, USA), as previously described [47]. Separation of the analytes was performed on a Poroshell 120 column (3.0 × 50 mm, 2.7 µm). The mobile phase composed of (A) methanol/acetonitrile/isopropanol (4/1/1) and (B) water, each containing 10 mM ammonium acetate and 0.2% formic acid, was delivered at a flow rate of 0.3 mL/min. Optimal separation of the analytes was achieved using the following elution gradient: 0–10 min, 60–100% A; 10–30 min, 100% A; and 30–35 min, 60% A. The transitions, fragmentor voltages, collision energies, and retention times are reported in Appendix A. Detection was performed after positive electrospray ionization at 300 °C at a flow rate of 10 L/min under 25 psi and 4000 V capillary voltage. The transitions, fragmentor voltages, and collision energies of the analytes available as standards were optimized using Agilent MassHunter Optimizer software. The parameters obtained were used for sphingolipids of the same class and closest molecular weight that were not available as standards. The chromatograms were analyzed using Agilent MassHunter quantitative analysis software. The sphingolipid concentrations were calculated by quadratic calibration using 1/x^2^ weighting factor. The linearity of the method of analysis for the 33 sphingolipids available as standards is provided in Appendix A. Good linearity was observed over a large range of concentrations. Good accuracy was observed over the range of concentrations assayed, as attested by the relative standard deviation, which was below 20% for each analyte (Appendix A).

### 4.6. Sphingolipids Extraction

The sphingolipids in the peritoneal fluid, peritoneal cells, plasma, and cells obtained from the spleen were extracted as previously described in [47] with slight modification. Briefly, physiological serum (0.9% NaCl) was added to 40 µL of cells and plasma and 120 µL of peritoneal fluid to a final volume of 160 µL. The samples were sonicated for 30 s in a Branson 2510 ultrasonic bath (Branson Ultrasonics Corporation, Danbury, CT, USA), and 600 µL of methanol/chloroform (2/1, *v/v*) and 10 µL of the mix of internal standard (commercial solution diluted four times in ethanol) were then added. In parallel, tubes containing only 10 µL of IS were prepared to assess the recovery. The tubes were then incubated overnight at 48 °C in a Memmert UM500 dry incubator (Memmert GmBH, Schwabach, Germany). The next day, 100 µL of 1M KOH solution was added to the samples; which were incubated for 2 h at 37 °C to cleave glycerophospholipids. Then, 10 µL of 50% acetic acid was added to neutralize the KOH, and the samples were homogenized and centrifuged for 15 min at 4500× *g*. The supernatants were collected, and the residues were extracted again with 600 µL of methanol/chloroform (2/1, *v/v*), centrifuged again, and then added to the first supernatant. The supernatants were evaporated on an SBH130D/3 block heater (Cole-Parmer, IL, USA) under air aspiration. Once completely dried, the samples were suspended in 200 µL of methanol, filtered (0.45 µm), transferred into vials, and 4–10 µL were injected into the UHPLC-MSMS system for sphingolipids analysis.

### 4.7. Sphingolipids Quantitation

The sphingolipid concentrations in the solution injected were determined using Agilent MassHunter quantitative analysis software from the calibration curves obtained for standards, as described in 4.5. The concentrations of sphingolipids that were not available as standards were calculated from the calibration curves obtained for the sphingolipids available as standards with the closest mass and abundance. Signal suppression and enhancement (SSE) calculated using the area method is reported for sphingolipids available as internal standards in Appendix A. Values of SSE outside 80–120% indicated that a matrix effect had occurred. Positive SSE was observed for d17:1P, d17:0P, and 18:1/12:0P in agreement with our previous results [47,68,79]. Intra-day and inter-day repeatability were evaluated by the RSD of the recovery measured on the IS. Good repeatability of the method of analysis was attested by low the RSD observed, which were below 20% except for 18:1/25:0, for which an RSD of 22% was observed. This RSD was in keeping with previous results observed in other matrices and was considered to be acceptable [47,68,79]. The final concentrations of sphingolipids in the peritoneal fluid, peritoneal cells, plasma, and cells of the spleen were corrected by the recovery measured for the corresponding IS as follows: d17:1 for d18:1, d17:0 for d18:0; d17:1P for d18:1P; d17:0P for d18:0P; 18:1/12:0 for 18:1/14:0, 18:1/16:0, 18:0/16:0, 18:1/18:1, and 18:1/18:0; 18:1/25:0 for all ceramides and all dihydroceramides for which the fatty acid chain length was C20 and above; SM18:1/12:0 for all sphingomyelins and all dihydrosphingomyelins; Glu18:1/12:0 for monohexosylceramides; Lac18:1/12:0 for lactosylceramides. No correction was used for 18:1/2:0, 18:0/2:0, GluSo, LysoSM, and LacSo.

### 4.8. Statistical Analysis

The statistical analyses of this study were performed with XLSTAT Biomed software (Addinsoft, Bordeaux, France). Differences in concentrations between groups were determined by one-way ANOVA after checking for homogeneity of variance (Hartley test). When a significant difference between groups was found (*p* < 0.05), an additional analysis of individual comparison of means was performed using Duncan’s test. Significantly different groups (*p* < 0.05) were indicated by different letters. The letter “A” is awarded to the group with the highest mean. Any two means having a common letter are not significantly different at the 5% level of significance. Correlations between sphingolipids were measured using the Pearson test. Significant correlations (*p* < 0.05) were indicated in bold. Sphingolipid concentrations in the different matrices analyzed were reported as means ± SD in the Tables and as means ± SE in the Figures.

## Figures and Tables

**Figure 1 ijms-24-02773-f001:**
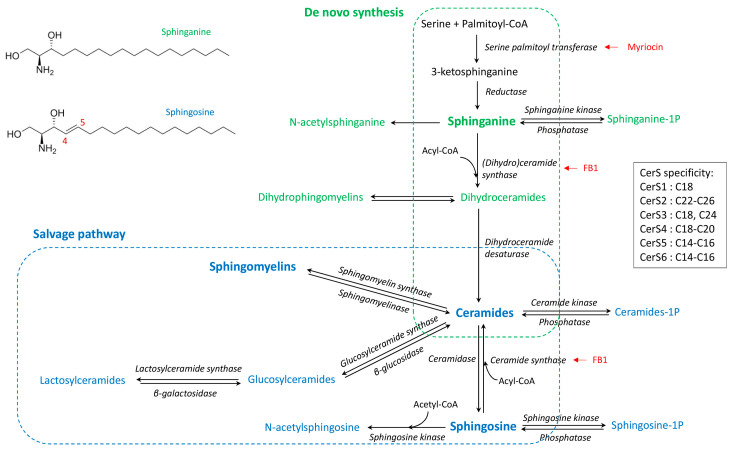
Simplified synthesis scheme of sphingolipids according to the de novo and salvage pathways, and the fatty acid chain length specificity of ceramide synthases (CerS). Dihydrosphingolipids formed with sphinganine are in green whereas sphingolipids formed with sphingosine are in blue. Known enzyme inhibitors of ceramides synthesis are in red [15].

**Figure 2 ijms-24-02773-f002:**
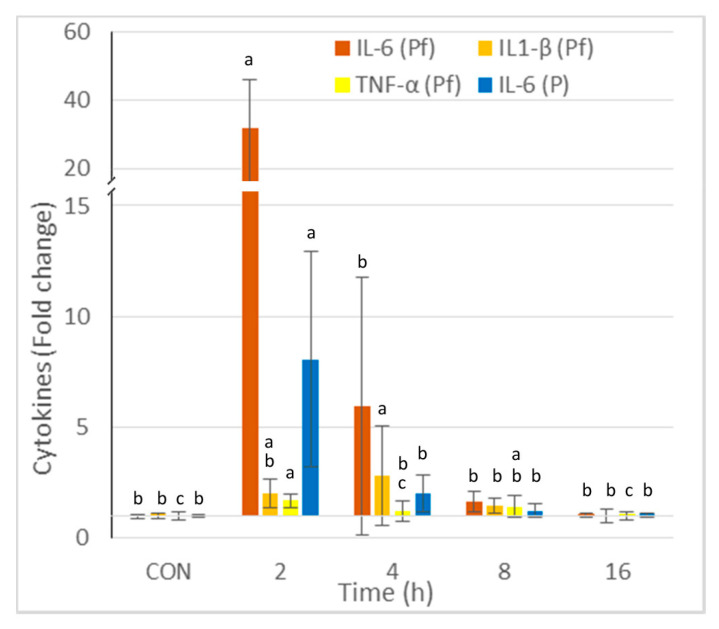
Effect of intraperitoneal administration of zymosan at a dose of 0.1 mg/mouse on the cytokines measured in the peritoneal fluid (Pf) and the plasma (P), expressed as fold-increase vs. the values measured in controls (CON) that received PBS only. Values are reported as means ± SD, *n* = 8 for the controls, and *n* = 4, for the mice treated at 2, 4, 8, and 16 h post-zymosan injection, respectively. Mean values of IL-6, IL1-β, and TNF-α in Pf, and IL-6 in P in CON were 27, 75, 51, and 758 pg/mL. Significant differences among groups were investigated using one way ANOVA. Statistically different groups (Duncan) were then identified using different apex letters (*p* < 0.05). Any two means having a common letter are not significantly different at the 5% level of significance.

**Figure 3 ijms-24-02773-f003:**
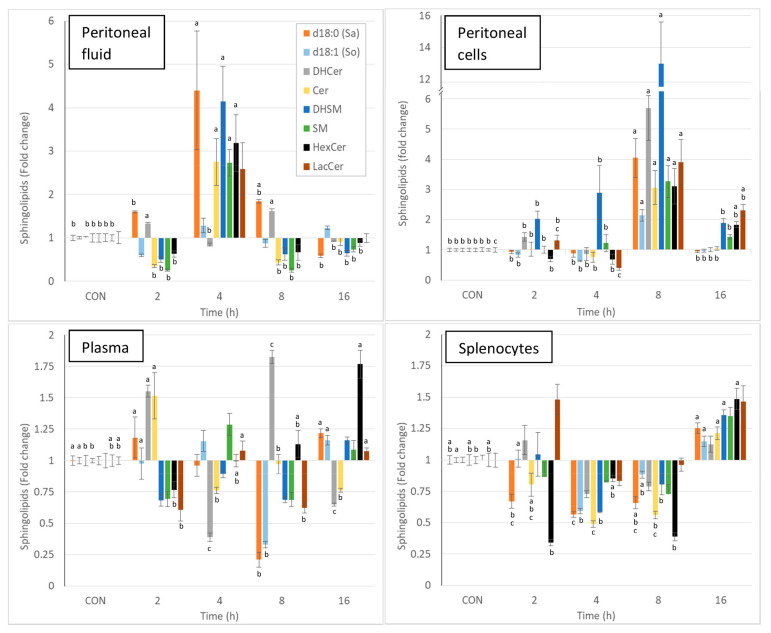
Effect of intraperitoneal administration of zymosan at a dose of 0.1 mg/mouse on the different classes of sphingolipids expressed as fold-increase vs. the values measured in controls (CON) that received PBS only. Values are reported as means ± SE, *n* = 16 for the controls and *n* = 4, 4, 4, and 8 for the mice treated at 2, 4, 8, and 16 h post-zymosan injection, respectively. Significant differences among groups were investigated using one way ANOVA. Statistically different groups (Duncan) were then identified using different apex letters (*p* < 0.05). Any two means having a common letter are not significantly different at the 5% level of significance.

**Figure 4 ijms-24-02773-f004:**
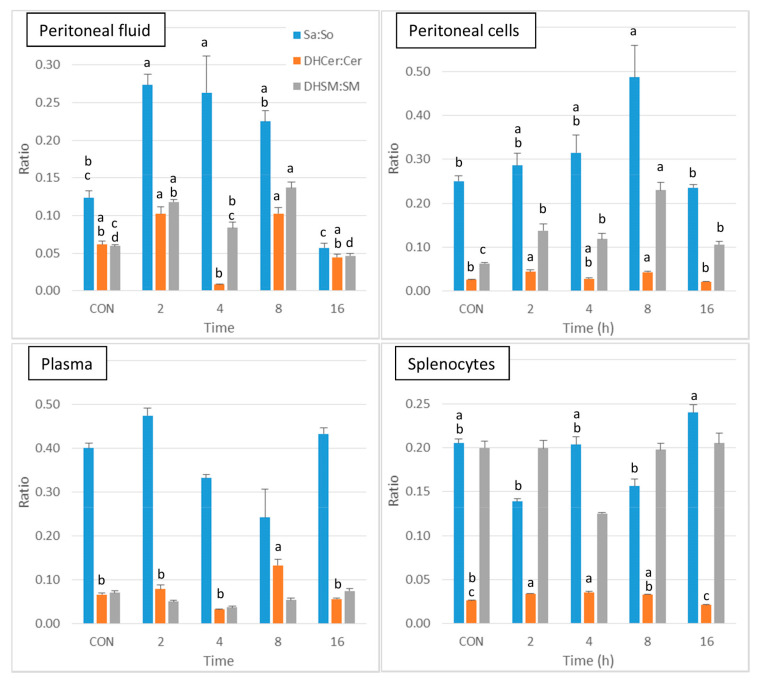
Sphingolipid ratios measured in control mice that received PBS (CON) and in mice after intraperitoneal administration of zymosan at a dose of 0.1 mg/mouse at 2, 4, 8, and 16 h post-zymosan injection. Sphinganine to sphingosine (Sa:So), dihydroceramides to ceramides (DHCer:Cer), and dihydrosphingomyelins to sphingomyelins (DHSM:SM) measured in the peritoneal fluid, peritoneal cells, plasma and the splenocytes. Values are reported as means ± SE, *n* = 16 for the controls and *n* = 4, 4, 4, and 8 for the mice treated at 2, 4, 8, and 16 h post-zymosan injection, respectively. Significant differences among groups were investigated using one way ANOVA. Statistically different groups (Duncan) were then identified using different apex letters (*p* < 0.05). Any two means having a common letter, are not significantly different at the 5% level of significance.

**Figure 5 ijms-24-02773-f005:**
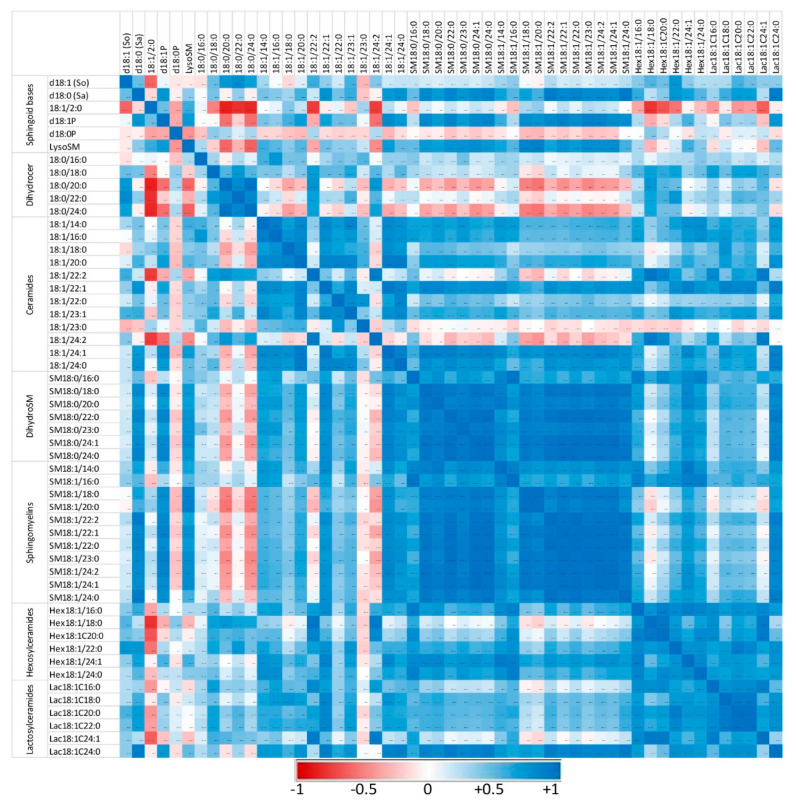
Correlation heatmap of sphingoid bases and sphingolipids with 14 to 24 carbon fatty acid chain lengths detected in the peritoneal cells of control mice that received PBS. Correlations are presented by sphingolipid class and by increasing size. The values of the Pearson coefficients of correlations observed among all sphingolipids assayed in this study are reported in Appendix A. Blue represents a positive correlation, red represents a negative correlation, and white represents no correlation.

**Figure 6 ijms-24-02773-f006:**
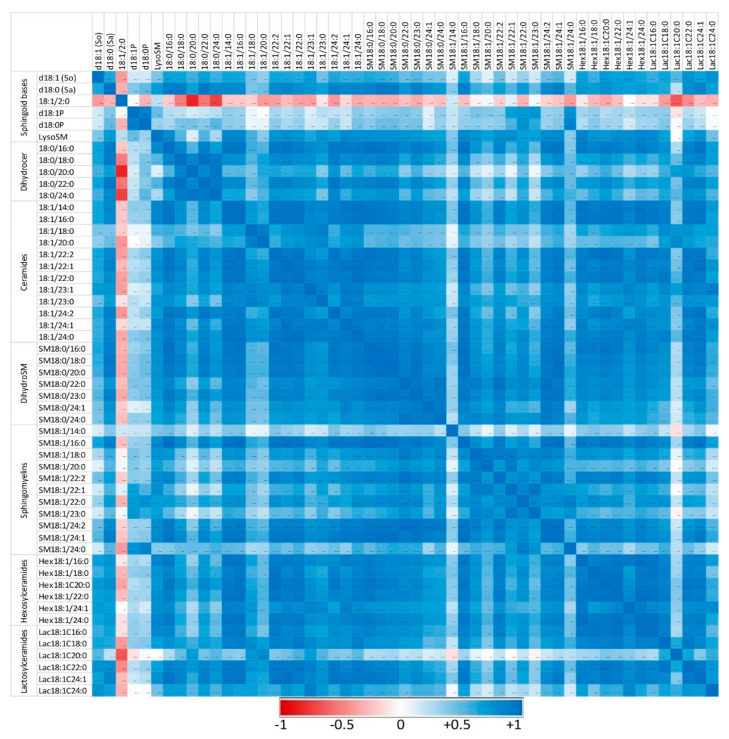
Correlation heatmap of sphingoid bases and sphingolipids with 14 to 24 carbon fatty acid chain lengths detected in the peritoneal cells of mice that received zymosan at a dose of 0.1 mg/mouse by the intraperitoneal route. Correlations are presented by sphingolipid class and by increasing size. The values of the Pearson coefficients of correlations observed among all sphingolipids assayed in this study are reported in Appendix A. Blue represents a positive correlation, red represents a negative correlation, and white represents no correlation.

**Table 1 ijms-24-02773-t001:** Total number and populations of cells obtained by lavage of the peritoneal cavity measured in control mice that received PBS and in mice injected with zymosan.

	Groups ^1^				
	CON	2	4	8	16
Peritoneal cells					
Total number ^2,3^	4 ± 3.4 ^B^	10 ± 8.7 ^AB^	1.9 ± 1.7 ^B^	17.8 ± 16 ^A^	10.9 ± 1.4 ^AB^
CD45^+ 3^	0.46 ± 0.4 ^B^	0.88 ± 0.8 ^B^	1.2 ± 1.1 ^B^	2.25 ± 2.7 ^B^	7.85 ± 3.2 ^A^
Ly6G^+ 4^	1.75 ± 2.5 ^B^	63.5 ± 60.7 ^B^	71.4 ± 57.7 ^B^	121 ± 147 ^B^	339 ± 212 ^A^
F4/80^+ high 4^	16.8 ± 19.1 ^AB^	0.47 ± 0.4 ^B^	0.19 ± 0.3 ^B^	0.93 ± 1.0 ^B^	30.6 ± 14.8 ^A^

^1^ Zymosan was injected at a dose of 0.1 mg/mouse; control mice (CON) received PBS only. The results are expressed as means ± SD, *n* = 12 for CON, and *n* = 4 for zymosan-treated mice at 2, 4, 8, and 16 h post-injection. ^2^ Corresponding to the total number of peritoneal cells recovered after lavage of the peritoneal cavity, ^3^ in 10^6^ cells, ^4^ in 10^4^ cells. Significant differences among groups were investigated using one way ANOVA. Statistically different groups (Duncan) were then identified using different superscript letters (*p* < 0.05). Any two means having a common letter are not significantly different at the 5% level of significance.

**Table 2 ijms-24-02773-t002:** Concentrations of sphinganine, sphingosine, dihydroceramides, ceramides, dihydrosphingomyelins, sphingomyelins, monohexosylceramides, and lactosylceramides measured in the peritoneal fluid, peritoneal cells, plasma, and the spleens of mice injected with zymosan or PBS.

	Groups ^1^				
	CON	2	4	8	16
Peritoneal fluid ^2^					
d18:0 (Sa)	5 ± 3 ^B^	7 ± 1 ^B^	20 ± 25 ^A^	8 ± 1 ^AB^	3 ± 2 ^B^
d18:1 (So)	46 ± 16	27 ± 6	59 ± 31	40 ± 16	56 ± 15
DHCer	27 ± 10 ^B^	35 ± 2 ^A^	21 ± 7 ^B^	43 ± 5 ^A^	24 ± 4 ^B^
Cer	1124 ± 1329 ^B^	391 ± 175 ^B^	3090 ± 2408 ^A^	484 ± 273 ^B^	1020 ± 833 ^B^
DHSM	432 ± 504 ^B^	216 ± 107 ^B^	1787 ± 1408 ^A^	270 ± 253 ^B^	278 ± 241 ^B^
SM	7150 ± 7743 ^B^	1804 ± 768 ^B^	19,499 ± 8714 ^A^	1787 ± 1278 ^B^	5248 ± 3149 ^B^
HexCer	182 ± 152 ^B^	115 ± 53 ^B^	581 ± 476 ^A^	122 ± 132 ^B^	160 ± 199 ^B^
LacCer	73 ± 120	ND	190 ± 178	ND	73 ± 59
Peritoneal cells ^3^					
d18:0 (Sa)	49 ± 35 ^B^	45 ± 11 ^B^	43 ± 25 ^B^	198 ± 127 ^A^	46 ± 17 ^B^
d18:1 (So)	212 ± 114 ^B^	177 ± 65 ^B^	135 ± 24 ^B^	454 ± 157 ^A^	205 ± 86 ^B^
DHCer	76 ± 56 ^B^	108 ± 43 ^B^	65 ± 63 ^B^	432 ± 324 ^A^	76 ± 41 ^B^
Cer	3252 ± 2053 ^B^	3327 ± 3049 ^B^	2483 ± 2161 ^B^	9967 ± 7321 ^A^	3408 ± 1568 ^B^
DHSM	1560 ± 1170 ^B^	3165 ± 1641 ^B^	4517 ± 5578 ^B^	20,256 ± 16,346 ^A^	2967 ± 1579 ^B^
SM	25,187 ± 19,968 ^B^	25,271 ± 11,547 ^B^	30,883 ± 28,742 ^B^	82,388 ± 53,526 ^A^	35,898 ± 14,463 ^B^
HexCer	1649 ± 923 ^B^	1152 ± 578 ^B^	1130 ± 1086 ^B^	5113 ± 3890 ^A^	3028 ± 1282 ^AB^
LacCer	440 ± 395 ^C^	577 ± 325 ^BC^	174 ± 115 ^C^	1715 ± 1354 ^A^	1017 ± 650 ^AB^
Plasma ^2^					
d18:0 (Sa)	58 ± 26 ^A^	68 ± 38 ^A^	55 ± 20 ^A^	12 ± 14 ^B^	70 ± 16 ^A^
d18:1 (So)	146 ± 47 ^A^	139 ± 71 ^A^	165 ± 47 ^A^	47 ± 14 ^B^	166 ± 43 ^A^
DHCer	223 ± 107 ^B^	346 ± 43 ^A^	87 ± 30 ^C^	407 ± 46 ^A^	145 ± 24 ^C^
Cer	3456 ± 808 ^B^	5235 ± 2547 ^A^	2631 ± 337 ^B^	3365 ± 1073 ^B^	2639 ± 420 ^B^
DHSMs	4309 ± 1762	2937 ± 771	3841 ± 472	2964 ± 403	5007 ± 970
SM	86,197 ± 59,296	59,731 ± 20,989	110,882 ± 30,101	59,311 ± 18,865	93,580 ± 52,821
HexCer	5838 ± 3263 ^AB^	4478 ± 1457 ^B^	5834 ± 1163 ^AB^	6605 ± 2513 ^AB^	10,319 ± 5265 ^A^
LacCer	261 ± 96 ^AB^	158 ± 94 ^B^	282 ± 80 ^A^	162 ± 43 ^B^	280 ± 62 ^A^
Splenocytes ^3^					
d18:0 (Sa)	33 ± 14 ^AB^	22 ± 8 ^BC^	19 ± 3 ^C^	22 ± 6 ^ABC^	42 ± 11 ^A^
d18:1 (So)	158 ± 38 ^A^	160 ± 43 ^A^	94 ± 14 ^B^	140 ± 17 ^AB^	181 ± 52 ^A^
DHCer	66 ± 18	77 ± 31	48 ± 7	52 ± 9	74 ± 34
Cer	2856 ± 1416 ^AB^	2298 ± 1048 ^ABC^	1394 ± 291 ^C^	1602 ± 346 ^BC^	3462 ± 1143 ^A^
DHSM	3620 ± 1258 ^B^	3779 ± 2507 ^AB^	2126 ± 96 ^B^	2910 ± 1178 ^B^	4908 ± 1225 ^A^
SM	20,669 ± 9843	17,914 ± 9224	16,989 ± 990	15,055 ± 6083	27,909 ± 11,610
HexCer	5569 ± 3575 ^AB^	1911 ± 582 ^B^	4755 ± 559 ^AB^	2160 ± 727 ^B^	8263 ± 3806 ^A^
LacCer	441 ± 291	654 ± 214	368 ± 65	425 ± 91	646 ± 454

^1^ Zymosan was injected intraperitoneally at a dose of 0.1 mg/mouse; control mice (CON) received PBS only. The results are expressed as means ± SD in ^2^ pmol sphingolipids/mL, and ^3^ pmol sphingolipids/mL of centrifuged cells obtained as described in the material and methods. *n* = 16 for CON and *n* = 4, 4, 4, and 8 for zymosan-treated mice at 2, 4, 8, and 16 h post-injection, respectively. Significant differences among groups were investigated using one way ANOVA. Statistically different groups (Duncan) were then identified using different superscript letters (*p* < 0.05). Any two means having a common letter are not significantly different at the 5% level of significance.

**Table 3 ijms-24-02773-t003:** Ceramides ratios measured in the peritoneal fluid, peritoneal cells, plasma, and the spleens of mice injected with zymosan or PBS.

	Groups ^1^				
	CON	2	4	8	16
Peritoneal fluid					
C16:C24	1.12 ± 0.56	0.79 ± 0.20	1.11 ± 0.22	1.03 ± 0.27	1.76 ± 1.44
C14–C18:C20–C26	0.66 ± 0.09	0.55 ± 0.13	0.77 ± 0.12	0.67 ± 0.15	0.98 ± 0.55
Peritoneal cells					
C16:C24	0.95 ± 0.37	0.68 ± 0.08	0.83 ± 0.30	1.14 ± 0.22	0.96 ± 0.22
C14–C18:C20–C26	0.71 ± 0.25	0.51 ± 0.06	0.62 ± 0.16	0.84 ± 0.17	0.22 ± 0.18
Plasma					
C16:C24	0.43 ± 0.32	0.45 ± 0.25	0.28 ± 0.10	0.24 ± 0.04	0.50 ± 0.27
C14–C18:C20–C26	0.16 ± 0.04	0.30 ± 0.16	0.18 ± 0.07	0.20 ± 0.03	0.29 ± 0.18
Spleen					
C16:C24	0.55 ± 0.23	0.94 ± 0.20	0.60 ± 0.06	0.96 ± 0.15	0.64 ± 0.29
C14–C18:C20–C26	0.47 ± 0.18	0.78 ± 0.17	0.53 ± 0.05	0.76 ± 0.11	0.68 ± 0.25

^1^ Zymosan was injected at a dose of 0.1 mg/mouse; control mice (CON) received PBS only. C16 to C24 ceramide ratio (C16:C24) corresponded to the ratio among 18:1/16:0 and the sum of 18:1/24:0, 18:1/24:1, and 18:1/24:2, and the C14–C18 to C20–C26 ceramide ratio (C14–C18:C20–C26) corresponded to the ratio between the sum of ceramides for which the fatty acid was of 14 to 18 carbons divided by the sum of ceramides for which the fatty acid was of 20 to 26 carbons measured in the peritoneal cells. Values are reported as means ± SD, *n* = 16 for the controls, and *n* = 4, 4, 4, and 8 for the mice treated at 2, 4, 8, and 16 h post-zymosan injection, respectively. No significant differences among groups were found using one way ANOVA (*p* > 0.05).

## Data Availability

The data presented in this study are available on request from the corresponding author.

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
