# Peer review of "Zymosan-Induced Murine Peritonitis Is Associated with an Increased Sphingolipid Synthesis without Changing the Long to Very Long Chain Ceramide Ratio"

_ijms, 2023, doi:10.3390/ijms24032773_

Round 1

Reviewer 1 Report

The ms by Alix Pierron et al reports, by a sphingolipidomic analysis, the variations of sphingolipids occurring after an i.p. zymosan injection in an animal model, along different times-points after the treatment. This study adds further important information on the role of this important biologically active lipids in a pathological condition, like peritonitis. The lipid analysis has been performed in peritoneal cells, in peritoneal fluid, in plasma and in the spleen, before and after the treatment. Albeit the huge amount of data reported, and the difficulty to systematically analyse time-dependent variations of individual molecules, the authors reach the paradigm of a general increase of de-novo synthesized sphingolipids and of a weak significance in their chain length.

The paper is well written, the introduction is clear, the methods are adequate and the conclusions are supported by the results.

Minor corrections should be done as follows:

Introduction. The authors report that "a low dose" to induce peritonitis in such a model corresponds to "up to 50mg/kg", but in their study they use a dramatically lower dose of 0.1 mg/mouse. The authors should explain why the choice of such a low dose, compared with that of other studies.

Table I. Please specify the meaning of the apex A, B, C, AB and BC, since not all the readers are confident with such a citation (this is true also for Table 2 and figures). Moreover, it is not reported to what it is referred the "total number of (peritoneal cells?)". Are they referred to a specific volume of the peritoneal fluid or what else?  Finally, the trend is ondulatory. Is there a reliable explanation for this? 

Fig.2. The CON bars are redundant, since they are all stated as = 100. Better is to delete them and express the values as fold-increase vs CON, instead of Sphingolipid %.

It is better to use the term monohexosylceramide, instead of hexosylceramide.

Reference 51 should be completed with the authors' names.

Author Response

The ms by Alix Pierron et al reports, by a sphingolipidomic analysis, the variations of sphingolipids occurring after an i.p. zymosan injection in an animal model, along different times-points after the treatment. This study adds further important information on the role of this important biologically active lipids in a pathological condition, like peritonitis. The lipid analysis has been performed in peritoneal cells, in peritoneal fluid, in plasma and in the spleen, before and after the treatment. Albeit the huge amount of data reported, and the difficulty to systematically analyse time-dependent variations of individual molecules, the authors reach the paradigm of a general increase of de-novo synthesized sphingolipids and of a weak significance in their chain length.

The paper is well written, the introduction is clear, the methods are adequate and the conclusions are supported by the results.

Thank you for your time in reviewing this manuscript and your positive comments on this work. Our responses are in italics and appear in the new version of the manuscript in revision mode. The suggested Figure changes have resulted in formatting changes to the text of the manuscript which complicate reading in revision mode. Therefore, to simplify the proofreading work, the text changes have been highlighted in yellow.

Minor corrections should be done as follows:

Introduction. The authors report that "a low dose" to induce peritonitis in such a model corresponds to "up to 50mg/kg", but in their study they use a dramatically lower dose of 0.1 mg/mouse. The authors should explain why the choice of such a low dose, compared with that of other studies.

Thank you for this Remark. The dose of 0.1 mg zymosan per mouse used in this study has already been reported to induce reversible inflammatory peritonitis (see for example Newson et al, 2014). We selected this dose in order to demonstrate the effect of zymosan on sphingolipids and to avoid any cytotoxic side effects. This point has been clarified in the revised version of the manuscript, Newson et al, 2014 has been added to the bibliography, the introduction and discussion have been modified accordingly. Zymosan doses quoted in the introduction are now presented in mg/mouse to simplify comparisons.

Table I. Please specify the meaning of the apex A, B, C, AB and BC, since not all the readers are confident with such a citation (this is true also for Table 2 and figures).

Thank you for your comment, the following explanatory sentence has been added to the legend of the relevant Figures and Tables: “Any two means having a common letter, are not significantly different at the 5% level of significance.” Materials and Methods section of the manuscript was completed as follow: “The letter "A" is awarded to the group with the highest mean. Any two means having a common letter, are not significantly different at the 5% level of significance.”

Moreover, it is not reported to what it is referred the "total number of (peritoneal cells?)". Are they referred to a specific volume of the peritoneal fluid or what else? 

This number corresponds to the total number of peritoneal cells recovered after washing the peritoneal cavity. This point was clarified in foot-note of Table 1.

Finally, the trend is ondulatory. Is there a reliable explanation for this? 

We have no explanation for the decrease in cell number observed at 4h, but it was not significant.

Fig.2. The CON bars are redundant, since they are all stated as = 100. Better is to delete them and express the values as fold-increase vs CON, instead of Sphingolipid %.

Thank you for this suggestion, by removing the CON bars as suggested it is no longer possible to present the statistical analysis by group. We have therefore presented the data as fold-increase vs CON, but kept the CON to show the statistics.

It is better to use the term monohexosylceramide, instead of hexosylceramide.

Thank you for this comment, hexosylceramide has been replaced by monhexosylceramide throughout the manuscript.

Reference 51 should be completed with the authors' names.

Thank you for this comment, this point was corrected in the new version of the manuscript.

Reviewer 2 Report

 Title: "Zymosan-induced murine peritonitis is associated with an in- 2

creased de novo sphingolipid synthesis without changing the long to very long chain ceramide ratio."

Summary: The manuscript under review is focused on measuring the changes in the sphingolipidome via targeted UHPLC-MS/MS analysis at different time points during the first 16 h post intraperitoneal administration of zymosan in mice. The authors used Zymozan at a lower dose to permit spontaneous recovery of peritonitis and avoid interference with sphingolipid production due to the cellular damage that could potentially occur at high doses. The data presented in the manuscript are robust, novel, and the changes in the sphingolipidome are extremely interesting. However, the presentation and interpretation of these remarkable and significant results need further improvement and some additional experiments to provide the currently lacking evidence in support of the conclusion that the observed changes are indeed a result of activation of the de novo SL synthesis vs. the salvage pathway for ceramide production. Below are a few suggestions for improvement.

Specific comments and points for improvement:

·      A significant difficulty in following and understanding the data in this manuscript is the fact that each subsection in the “Results” section describes a vast amount of data stating the observed changes without providing any interpretation or explanation about the authors' reasoning as to why they did these analyses and how the results led them to the next set of analyses. I suggest adding 1-2 sentences at the end of each subsection summarizing the observations and helping the readers understand "So, what, what these changes suggest?". I believe this will significantly improve the quality of the manuscript and provide an interesting story rather than overwhelm the reader with numbers.

·      Figure 1, I believe, contains an error. "Salvage synthesis" should be "Salvage pathway" as this is a terminology explicitly used for the existing pathways for the synthesis of ceramide from sphingosine or the recycling of complex SLs (SM and different glycol-SLs, cerebrosides)- see references here: https://pubmed-ncbi-nlm-nih-gov.icom.idm.oclc.org/18191382/

·      Figure 1 could be improved by adding the respective enzyme inhibitors discussed by the authors in the introduction section, possibly in a contrast color (red or orange).

·      Tables 1 and 2 are missing labels – the numbers on the top need a legend and identification. A title on the top of each table would be excellent, too, and easier to follow. Small superscript letters (A, B) must be added to the legend with the respective explanations. Also, specify the type of ANOVA used in each figure/table legend as per the Materials and Methods section with details about the differences indicated in the respective figure/table.

·      Figure 2 could be improved by adding a title to each panel, identifying the tissue/cell type. In addition, the changes in panel B are difficult to see due to one single SL changing 12-fold. Changing the scale to 600% with a break in the scale and the individual SL species will help more clearly visualize the changes in peritoneal cells compared to the changes in the peritoneal fluid. Furthermore, since the data in this figure are represented as % change vs. the control, the controls (always the same) could be omitted from the figure, and the scale changed to include negative values for the decreases observed. This will make the data in the panels more "reader-friendly." Fold change instead of % change might be a more appropriate manner to represent these results. Similarly to other figures and tables, "a,b,c, etc." statistical changes need to be explained in the legend.

·      For Figure 3, please follow the suggestions to add a title to each panel and an explanation for the statistical changes abbreviations to the legend.

·      A significant problem with the data is the normalization of the mass spec data. For peritoneal fluid and plasma, pmol/ml is an acceptable normalization. However, it is unclear what the "ml" refers to for peritoneal cells and spleen tissue. Is it an ml of tissue/cell homogenate or something else? For these samples, more appropriate normalization factors are ether cell number, mg of total protein or DNA, mg of tissue, etc. I believe using the correct normalization for peritoneal cells vs. fluid will improve the manuscript's quality and help the authors interpret their results, which is the weakest point of this manuscript.

·      After the summary of the results in Figure 2, the authors suddenly decided to continue only with peritoneal fluid and cells without explaining why plasma and spleen are no longer included. Please provide a reasonable explanation.

·      Cytokines data will look much better if represented as bar graphs instead of numbers in a table. Since one of the initial statements of the authors is that they are comparing local vs. systemic inflammation and further include measurements of SLs in 4 different bodily constituents, it is necessary to add a figure with the changes of the same cytokines in plasma. This will provide a clearer picture of the possible origin of the observed differences.

·      The two heatmaps shown in Figures 4 and 5 are impossible to read. Perhaps adding a small table to each one with the most noticeable changes in SL species observed (fold increased and fold decreased) will significantly improve the presentation of results.

·      The biggest challenge with this study is that it is simply observational. Because of the vast complexity of the SL metabolism, the clear compartmentalization of different enzymes in different subcellular compartments, and the tissue-specific expression of many other enzymes, it is difficult, if impossible, to make any conclusions based solely on changes in certain SL species. Nevertheless, based on the reported changes, I agree with the authors that the most likely pathway affected by the zymosan administration in the peritoneal cell is the de novo pathway for SL synthesis. However, they must provide at least one additional piece of evidence to support this suggestion, such as enzyme activity assays for any of the enzymes of the de novo synthesis (SPT activity, Cer synthase, SM synthase) and the turnover pathway (different SMases, ceramidase, Sph-kinases).

Author Response

Summary: The manuscript under review is focused on measuring the changes in the sphingolipidome via targeted UHPLC-MS/MS analysis at different time points during the first 16 h post intraperitoneal administration of zymosan in mice. The authors used Zymozan at a lower dose to permit spontaneous recovery of peritonitis and avoid interference with sphingolipid production due to the cellular damage that could potentially occur at high doses. The data presented in the manuscript are robust, novel, and the changes in the sphingolipidome are extremely interesting. However, the presentation and interpretation of these remarkable and significant results need further improvement and some additional experiments to provide the currently lacking evidence in support of the conclusion that the observed changes are indeed a result of activation of the de novo SL synthesis vs. the salvage pathway for ceramide production. Below are a few suggestions for improvement.

Thank you for your time in reviewing this manuscript and your positive comments on this work. Our responses are in italics and appear in the new version of the manuscript in revision mode. The suggested Figure changes have resulted in formatting changes to the text of the manuscript which complicate reading in revision mode. Therefore, to simplify the proofreading work, the text changes have been highlighted in yellow.

Specific comments and points for improvement:

  • A significant difficulty in following and understanding the data in this manuscript is the fact that each subsection in the “Results” section describes a vast amount of data stating the observed changes without providing any interpretation or explanation about the authors' reasoning as to why they did these analyses and how the results led them to the next set of analyses. I suggest adding 1-2 sentences at the end of each subsection summarizing the observations and helping the readers understand "So, what, what these changes suggest?". I believe this will significantly improve the quality of the manuscript and provide an interesting story rather than overwhelm the reader with numbers.

Thanks to this suggestion, we have clarified the reason for the analyses carried out in each of the sub-sections of the revised version of the manuscript at the beginning or at the end.

2.1. Peritoneal cells, splenocytes, and cytokines

This sentence was added at the beginning of the subsection: « Measurements of cells and cytokines were performed to confirm that the dose of zymosan administered was responsible for a low grade non-infectious inflammatory peritonitis. ».

2.2. Sphingolipids by class

This sentence was added at the beginning of the subsection: « As the effects on sphingolipid metabolism are complex, a first analysis was carried out by classes. ».

These two sentences were added at the middle of the subsection: « Thus, the analysis of the different sphingolipid classes reveals that the local effects of zymosan are much stronger than its systemic effects. As all classes of sphingolipids were significantly altered after zymosan administration (Table 2), a further presentation of the results was carried out to compare the effects on sphinganine-based sphingolipids and on sphingosine-based sphingolipids. ».

This sentence was added at the end of the subsection: « Comparison of the Sa:So, DHCer:Cer, and DHSM:SM ratios shows that the effects of zymosan are more pronounced on sphinganine-containing sphingolipids than on sphingosine-containing ones.».

2.3. Sphingolipids according to the chain length of the fatty acid

Three sentence were added at the beginning of the subsection: « The comparison of the effects of zymosan on different classes of sphingolipids does not allow to determine whether different sphingolipids within the same class are affected in the same way. This information is nevertheless essential given the role of fatty acid chain length on sphingolipid. Given the number of sphingolipids measured, it is difficult to examine the effect of zymosan on an analyte by analyte basis. Ceramide ratios and sphingolipid correlations were therefore performed to reveal whether zymosan varied the respective proportions of sphingolipids ».

This sentence was added before Figures showing the correlations among sphingolipids: “The correlations between the different sphingolipids were also explored to reveal whether the administration of zymosan changed the respective ratios between compounds.”.

The last sentences of this sub-section have been reworded as follows: « Therefore, the effects of zymosan on sphingolipids are similar regardless of the length of the fatty acid incorporated in their synthesis. Analysis of the correlations observed be-tween the different analytes measured also revealed that zymosan administration induced a strong positive correlation between dihydroceramides and other sphingolipids, which was not observed in control mice. ».

  • Figure 1, I believe, contains an error. "Salvage synthesis" should be "Salvage pathway" as this is a terminology explicitly used for the existing pathways for the synthesis of ceramide from sphingosine or the recycling of complex SLs (SM and different glycol-SLs, cerebrosides)- see references here: https://pubmed-ncbi-nlm-nih-gov.icom.idm.oclc.org/18191382/
  • Figure 1 could be improved by adding the respective enzyme inhibitors discussed by the authors in the introduction section, possibly in a contrast color (red or orange).

Thank you for these comments, Figure 1 has been modified as suggested.

  • Tables 1 and 2 are missing labels – the numbers on the top need a legend and identification. A title on the top of each table would be excellent, too, and easier to follow. Small superscript letters (A, B) must be added to the legend with the respective explanations. Also, specify the type of ANOVA used in each figure/table legend as per the Materials and Methods section with details about the differences indicated in the respective figure/table.

Thank you for these comments, Tables and Figures legend have been modified as suggested. The following explanatory sentence has been added to the legend of the relevant Figures and Tables: “Any two means having a common letter, are not significantly different at the 5% level of significance.” Materials and Methods section of the manuscript was completed as follow: “The letter "A" is awarded to the group with the highest mean. Any two means having a common letter, are not significantly different at the 5% level of significance.”

We do not understand the comment about “A title on the top of each table” as title are already presented for each Table.

  • Figure 2 could be improved by adding a title to each panel, identifying the tissue/cell type. In addition, the changes in panel B are difficult to see due to one single SL changing 12-fold. Changing the scale to 600% with a break in the scale and the individual SL species will help more clearly visualize the changes in peritoneal cells compared to the changes in the peritoneal fluid. Furthermore, since the data in this figure are represented as % change vs. the control, the controls (always the same) could be omitted from the figure, and the scale changed to include negative values for the decreases observed. This will make the data in the panels more "reader-friendly." Fold change instead of % change might be a more appropriate manner to represent these results. Similarly to other figures and tables, "a,b,c, etc." statistical changes need to be explained in the legend.
  • For Figure 3, please follow the suggestions to add a title to each panel and an explanation for the statistical changes abbreviations to the legend.

Thank you for this suggestion, we have modified the Figures as suggested. We have kept the CON that is necessary to present the statistical analysis by group. Statistical changes have been better explained in the legend.

  • A significant problem with the data is the normalization of the mass spec data. For peritoneal fluid and plasma, pmol/ml is an acceptable normalization. However, it is unclear what the "ml" refers to for peritoneal cells and spleen tissue. Is it an ml of tissue/cell homogenate or something else? For these samples, more appropriate normalization factors are ether cell number, mg of total protein or DNA, mg of tissue, etc. I believe using the correct normalization for peritoneal cells vs. fluid will improve the manuscript's quality and help the authors interpret their results, which is the weakest point of this manuscript.

Thank you for this comment. We agree that the normalisation of the results is a key point in their correct interpretation. It is why 10 internal standards that are representative of each class of sphingolipids are added to each sample prior to analysis and that the results are presented corrected by the recovery observed on the different internal standards. Also, it is why sphingolipids analysis in cells were always performed on 40µL of centrifuged cells, as indicated in the material and methods section of the manuscript. This information was added in a footnote in Table to clarify.

Expression of results in mL/centrifuged cells rather than in millions of cells allows the comparison of the concentrations of sphingolipids in the different matrices. Moreover, in our expertise on the measure of the sphingolipidome in brain, lung, liver, kidney, heart, muscle, plasma, and milk, expressing sphingolipids contents per amount of tissue sampled is the best way to avoid biases due to the conversion of results using another variable.

In addition, the effects of zymosan were measured on sphingolipid ratios. This approach is probably the best way to avoid false interpretations that would be made on the total amounts alone. Ratios of sphingolipids, and measurement of the sphinganine:sphingosine ratio, is widely used as biomarker to reveal exposure to fumonisins, which are mycotoxins that contaminate human and animal food [ref 15 of the manuscript, for a review]. Recent studies also suggest an interest of the ratio between long chain ceramides and very long chain ceramides to characterise the cardiotoxic risk in humans [ref 38 and 39 of the manuscript].

  • After the summary of the results in Figure 2, the authors suddenly decided to continue only with peritoneal fluid and cells without explaining why plasma and spleen are no longer included. Please provide a reasonable explanation.

Many thanks you for this remark, we focused our analysis on peritoneal fluid and peritoneal cells because the effects of zymosan were strongest in these matrices. Nevertheless we understand that the lack of presentation of the analysis of the effects in plasma and spleen could be confusing, and these analysis were added in the revised version of the manuscript. Figure 3 has been redrawn to focus on dihydrosphingolipid:sphingolipid ratios, and Table 3 has been added to focus on C16:C24 and C14-C18:C20-C26 ratios. In order not to overload the manuscript, the SM:Cer, HexCer:Cer, and LacCer:Cer ratios have been presented for the 4 matrices in Figure S1. The text has been modified accordingly. These new results strengthen the interpretation of the effects of zymosan.

  • Cytokines data will look much better if represented as bar graphs instead of numbers in a table. Since one of the initial statements of the authors is that they are comparing local vs. systemic inflammation and further include measurements of SLs in 4 different bodily constituents, it is necessary to add a figure with the changes of the same cytokines in plasma. This will provide a clearer picture of the possible origin of the observed differences.

Thank you for this comment. The cytokine data are now presented in Figure 2 as suggested. New results for IL-6 in plasma have been added to the revised version of the manuscript. IL-6 was chosen as it had the highest variation in the peritoneal fluid and it was not possible to measure all the cytokines due to the volume of plasma remaining at our disposal. These new results corroborate the results obtained on sphingolipids.

The following sentence was added to the discussion: The effects of zymosan on IL-6 concentrations were more marked in the peritoneal fluid than the plasma.

  • The two heatmaps shown in Figures 4 and 5 are impossible to read. Perhaps adding a small table to each one with the most noticeable changes in SL species observed (fold increased and fold decreased) will significantly improve the presentation of results.

Thank you for this comment. We have clarified at the beginning of this section that: “The comparison of the effects of zymosan on different classes of sphingolipids does not allow to determine whether different sphingolipids within the same class are affected in the same way. This information is nevertheless essential given the role of fatty acid chain length on sphingolipid. Given the number of sphingolipids measured, it is difficult to examine the effect of zymosan on an analyte by analyte basis. Ceramide ratios and sphingolipid correlations were therefore performed to reveal whether zymosan varied the respective proportions of sphingolipids.”

The aim of these figures is therefore to visually reveal whether the correlations between sphingolipids change after the administration of zymosan. This representation allows the rapid identification of possible specific effects within a large number of sphingolipids. It reveals in the case of zymosan that the correlations among sphingolipids are similar regardless of the fatty acid incorporated. Also, this presentation shows positive correlations between dihydroceramides and other sphingolipid regardless of the size of the fatty acid in mice given zymosan, but not in controls. This observation is therefore complementary to the analysis of the ratios between dihydrosphingolipids and sphingolipids, and suggests that the increase in synthesis occurred whatever chain length of the fatty acid.

  • The biggest challenge with this study is that it is simply observational. Because of the vast complexity of the SL metabolism, the clear compartmentalization of different enzymes in different subcellular compartments, and the tissue-specific expression of many other enzymes, it is difficult, if impossible, to make any conclusions based solely on changes in certain SL species. Nevertheless, based on the reported changes, I agree with the authors that the most likely pathway affected by the zymosan administration in the peritoneal cell is the de novo pathway for SL synthesis. However, they must provide at least one additional piece of evidence to support this suggestion, such as enzyme activity assays for any of the enzymes of the de novo synthesis (SPT activity, Cer synthase, SM synthase) and the turnover pathway (different SMases, ceramidase, Sph-kinases).

Thank you for this comment. As you point out the increase in de novo synthesis is the only one that can explain all the changes observed in this study. To our knowledge there is no study in any field that shows an increase in together sphinganine, dihydroceramides, dihydrosphingomyelin and glycosylceramides by any other way than increased de novo synthesis. Furthermore, if an increase in ceramide and sphingosine levels could be related to an activation of the salvage pathway, these effects should be associated with a decrease in sphingomyelins or glycosylceramides. In contrary, these compounds were increased. Additionally, if we take into account the ratio between ceramides and sphingomyelins, it is an increase of more than 10 times of ceramides production that was necessary to reach the sphingomyelin levels measured in the peritoneal cells at 8 hours post-injection. However, we remained cautious in our conclusions and said that the study “suggests” an increase in de novo sphingolipid synthesis. Also, and to avoid any confusion, we removed “de novo” from the title of the manuscript, and have added in the conclusion that further analysis at protein level are required to clarify the mechanisms involved in the effect observed.

Unfortunately, analyses at the protein level will not be possible in this work as the experimental design was not made for this. No studies to date report the effects of zymosan on the sphingolipidome, nor are these effects described for any TLR2/Dectin agonist, either in vivo or in vitro. The aim of this study was therefore to describe the effects of a low dose of zymosan on the sphingolipidome, both locally and systemically. The changes observed suggests an activation of de novo synthesis. This result was not expected when we set up the experimental protocol, and it was not possible to predict what effect, if any, the post-injection delay would have on the sphingolipidome. Also, even local effects were expected to be stronger that systemic effect in terms of inflammatory response, we were not sure that it would be the same in terms of change in the sphingolipidome. Indeed, and to our knowledge, no other study to date has compared the local and systemic effects of an infectious challenge on the sphingolipidome. So we agree that further works are necessary at the enzymatic/mRNA level, and they could also include analysis of the effect of specific enzyme inhibitors.

Round 2

Reviewer 2 Report

I agree with the significantly improved version and the authors' responses.